# Transcriptomic Profile of Tef (*Eragrostis tef*) in Response to Drought

**DOI:** 10.3390/plants13213086

**Published:** 2024-11-02

**Authors:** Lorena Ramirez-Gonzales, Gina Cannarozzi, Abiel Rindisbacher, Lea Jäggi, Regula Schneider, Annett Weichert, Sonia Plaza-Wüthrich, Solomon Chanyalew, Kebebew Assefa, Zerihun Tadele

**Affiliations:** 1Institute of Plant Sciences, University of Bern, 3013 Bern, Switzerland; lorena.ramirez@unibe.ch (L.R.-G.); gina@cannarozzi.com (G.C.); abiel.rindisbacher@unibe.ch (A.R.); lea.jaeggi@unibe.ch (L.J.); regisch@bluewin.ch (R.S.); annett.weichert@gmail.com (A.W.); plaza_sonia@hotmail.com (S.P.-W.); 2Ethiopian Institute of Agricultural Research, Addis Ababa P.O. Box 2003, Ethiopia; solchk2@gmail.com (S.C.); kebebewassefa1@gmail.com (K.A.)

**Keywords:** abiotic stress, drought, *Eragrostis tef*, differentially expressed genes (DEG), RNA-Seq, transcriptome

## Abstract

The threat to world food security posed by drought is ever increasing. Tef [*Eragrostis tef* (Zucc.) Trotter] is an allotetraploid cereal crop that is a staple food for a large population in the Horn of Africa. While the grain of tef provides quality food for humans, its straw is the most palatable and nutritious feed for livestock. In addition, the tef plant is resilient to several biotic and abiotic stresses, especially to drought, making it an ideal candidate to study the molecular mechanisms conferring these properties. The transcriptome expression of tef leaf collected from plants grown under drought conditions was profiled using RNA-Seq and key genes were verified using RT-qPCR. This study revealed that tef exhibits a complex molecular network involving membrane receptors and transcription factors that regulate drought responses. We identified target genes related to hormones like ABA, auxin, and brassinosteroids and genes involved in antioxidant activity. The findings were compared to physiological measurements such as changes in stomatal conductance and contents of proline, chlorophyll and carotenoid. The insights gained from this work could play vital role in enhancing drought tolerance in other economically important cereals such as maize and rice.

## 1. Introduction

Drought is a major factor affecting crop productivity worldwide, particularly in developing countries. In Ethiopia, the ancient cereal tef [*Eragrostis tef* (Zucc.) Trotter] is considered to be a drought-tolerant crop compared to other commercial cereals such as wheat and maize. However, during periods of water shortage due to low rainfall in some tef growing areas in Ethiopia, tef is still affected, which can cause a yield loss of up to 40% [1]. Thus, understanding the genetic basis of the plant response to abiotic constraints is fundamental for the development of stress-resilient tef varieties—a knowledge that in the future can be extrapolated to improve water use efficiency in commercial cereals.

The allotetraploid grass tef (2*n* = 4x = 40), a member of the family *Poaceae* and the subfamily *Chloridoidae*, is a staple food to over 70 million people in Ethiopia. It is the only member of the *Eragrostis* Genus used for human consumption, although several species are used as livestock fodder [2]. Tef grows in a wide variety of agro-ecological conditions, ranging from semi-arid to high-rainfall [3]. In addition, tef has a significant amount of diversity [4], is resistant to several biotic and abiotic stresses, has a desirable nutritional content [5], and is gluten-free [6].

Drought stress causes alterations in plant morphology, physiology, and biochemistry that adversely affect the productivity and growth of the plant. It also causes an accumulation of abscisic acid (ABA)—a phytohormone that triggers a molecular mechanism to cope with drought stress. This is achieved by regulating stomatal conductance, plant growth, and development as well as reproductive processes [7]. These molecular mechanisms can be grouped under three categories: (i) signaling and transcriptional control; (ii) protection of membranes and proteins, such as heat shock proteins, osmoprotectants (e.g., proline), and free-radical scavengers; and (iii) water and ion transport and uptake mediated by for example aquaporins or ion transporters reviewed in [8]. Moreover, plants synthesize different types of compounds to overcome drought stress including amino acids [9], sugars [10], and lipids [11].

Although tef is considered to be an orphan crop due to the dearth of scientific research performed on the crop [12], a limited number of studies have been realized, including the whole genome sequencing of two genotypes [13,14]. In addition, studies have been conducted to investigate the differential expression of genes under flooding or waterlogging [15] as well as the expression of microRNAs under drought stress [16], calcium deficiency [17], and the characterization of repetitive sequences [18].

A proteomics study in tef conducted under drought conditions using iTRAQ quantitative mass spectrometry revealed 211 differentially expressed proteins that are mainly grouped under ROS (reactive oxygen species)-production processes and cell wall modification [19]. Furthermore, a metabolite study under drought conditions showed a higher accumulation of metabolites associated with drought tolerance as sugar metabolism and amino acids [20]. A recent review on tef omics provides a detailed account of relevant studies made on the crop [21].

In general, the high-throughput sequencing of an organism using diverse omics techniques provides information vital to understanding the mechanism of stress tolerance in general and drought tolerance in particular. For instance, the findings from the RNA-Seq can identify differentially expressed genes under diverse moisture regimes, including drought. Earlier RNA-Seq studies on cereal crops under drought conditions revealed differentially regulated genes in wheat [22], rice [23], foxtail millet [24], and sorghum [25].

In the first study, a possible drought tolerance mechanism was discussed using two drought-tolerant wheat genotypes; some of the genes identified were involved in flavonoid biosynthesis and fructan biosynthesis in starch and sucrose metabolism [22]. Regarding the study on rice, they found that drought tolerance was enhanced during root development by increasing the levels of catalase and ascorbate peroxidase enzymes that are negatively regulated by phytochrome B [23]. In foxtail millet, short drought stress exposure (24 h) showed changes in key genes related to chlorophyll synthesis, proline synthesis, and other pathways [24]. Finally, a RNA-Seq study performed in both drought-tolerant and drought-sensitive sorghum genotypes found that the sensitive one had a lower expression of genes related to abiotic stimulus, oxidoreductase activity, and response to stress, whereas the drought-tolerant genotype showed an increased in the expression levels of cuticular synthesis genes [25].

Although tef is extensively cultivated in semi-arid areas in the Horn of Africa where moisture scarcity is the major constraint and the crop serves as a staple food for millions of people, RNA-Seq studies involving drought conditions have not yet been conducted. Hence, differentially expressed genes were not known for this drought-tolerant crop. Among species in the *Eragrostis* genus, *E. capensis* was found to be drought-intolerant, while tef was moderately drought-tolerant and *E. curvula* was drought-tolerant [26]. The current study investigates differentially expressed genes (DEGs) in tef plants exposed to moisture scarcity and validates candidate genes using RT-qPCR.

## 2. Results

### 2.1. Effect of Drought on Physiological and Morphological Parameters

To determine the effect of moisture scarcity on plant physiology, measurements were made on key parameters, including chlorophyll content, carotenoid content, and stomatal conductance. For chlorophyll a and cartenoid contents, no significant differences were found between well-watered and drought conditions (Figure 1A,D). However, significant differences were obtained for chlorophyll b (*p* < 0.05), total chlorophyll (*p* < 0.05), and stomatal conductance (*p* < 0.001) (Figure 1B,C,E). As expected, the values of all tested physiological parameters were lower in drought-treated plants compared to well-watered ones, although the differences were insignificant for some parameters. It is important to note that the stomatal conductance of the leaf was substantially reduced under drought where it was only 50% of the average value of well-watered plants. These experiments showed that the *Tsedey* genotype suffers at least at a modest level after nine days of water withholding. The effect of the 9-day water withholding is visually observed in Figure 1F where plants under moisture scarcity showed obvious drought-related symptoms such as wilting. We thus concluded that these samples from well-watered and drought treatments can be used for the RNA-Seq analysis.

### 2.2. RNA-Seq Analysis and Differentially Expressed Genes (DEGs)

Four libraries were used in this study. Two controls from plants with optimal water conditions (GNY1, GNY10) and other two from drought treated plants (GNY2, GNY11). The percentages of reads uniquely mapped to one position in the genome per each library were 75.9% for GNY1, 66.93% for GNY2, 87.11% for GNY11, and 88.14% for GNY10. On the other hand, reads mapped to multiple loci per library were 12.75% for GNY1, 12.21% for GNY2, 10.8% for GNY11, and 10.06% for GNY10. However, no read from all libraries was unmapped. A background set of transcripts was defined by mapping the reads onto the set of 66,287 genes predicted from the tef *Dabbi* genome assembly [14]. For the differential expression analysis, count tables were generated using HTSeq [27], and expression analysis was conducted using DESeq2 [28]. These procedures identified 773 significantly downregulated and 671 significantly upregulated genes under drought conditions (Appendix A).

### 2.3. GO Classification and Enrichment Analysis

Using the R package topGO and the GO term annotation from the reference *Dabbi* tef genotype, DEGs were annotated with functions from three classes: biological processes (BP), cellular components (CCs), and molecular function (MF). For upregulated genes, the most significant enrichment in GO terms in BP were “hormone-mediated signaling pathway”, and “cellular response to hormone stimulus” (Figure 2A), while for MF, the top function was “protein kinase activity” (Figure 2C). Similarly, the most enriched CCs were “protein-containing complex” and “apoplast” (Figure 2E). Regarding downregulated genes, the most downregulated BPs were “dephosphorylation” and “protein dephosphorylation” (Figure 2B), while for MF, they were “phosphoprotein phosphatase activity” and “protein serine/threonine phosphatase activity” (Figure 2D). The most downregulated CC was “cytoskeleton” (Figure 2F).

### 2.4. Pathway Analysis of DEGs

The genes were annotated by Mercator, a web application tailored to the functional annotation of plants [29], and then these annotations (Appendix A), together with the RNA-Seq expression values, were used as input in the MapMan software, which constructed visualizations of metabolic and stress-related pathways [30]. As shown in Appendix A, 198 genes were upregulated, while 282 genes were downregulated. The pathway overview of biotic and abiotic stress-related genes is shown in Appendix A. The highest number of upregulated genes (red) belong to ABA regulation, heat shock proteins, and abiotic stress response, whereas the majority of downregulated genes (blue) are involved in cell walls, proteolysis, signaling, peroxidase regulation, abiotic stress, and secondary metabolites (Appendix A). This is in agreement with the GO functional annotation of biological process, where the upregulated genes were enriched with the “hormone-mediated signaling pathway” that corresponds to the high number of upregulated genes in the ABA regulation pathway obtained by Mercator (Figure 2C). In the case of the downregulated genes, the GO term “protein phosphatase activity” can also be related to the signaling pathway found with the Mercator tool (Figure 2D).

Regarding metabolism, 79 upregulated and 102 downregulated genes were mapped (Appendix A). Similarly, as in the case with biotic/abiotic stress-related genes, those downregulated in metabolism were also involved in cell wall regulation (Appendix A). Other upregulated metabolic processes were related to lipids, terpenes, flavonoid metabolism, glycolysis, and light reactions (Appendix A). In contrast, downregulated genes were found mostly in minor CHO (raffinose metabolism) and starch and sucrose metabolisms (Appendix A).

Interestingly, several genes encoding receptor-like/Pelle kinases (RLKs)- signaling proteins that regulate developmental processes and stress responses - were downregulated (Appendix A). The family with the highest number of members differentially expressed belong to the leucine-rich repeats (LRR) family, followed by members of WAKs (Wall-associated kinases) family and L-Lectin receptors. Some genes encoding proteins belonging to the LRR, DUF26, WAK, and Thaumatin family receptors were also upregulated. The complete list of genes involved in signaling is shown in Appendix A.

### 2.5. Regulation of Hormone Signaling and Genes Directly Involved in Abiotic Stress

Plants’ early response to drought stress is linked to the synthesis of plant hormones. We found some genes involved in abscisic acid hormone regulation under drought stress (Appendix A). Here, genes directly involved in the metabolic pathway of ABA biosynthesis such as *NCED* (*9 cis Epoxycarotenoid Dioxygenease NCED Chloroplast*), *Abscisic Aldehyde Oxidase (AAO)*, and *GT (UDP Glycosyltransferase)* were upregulated under drought stress (Figure 3). Moreover, we found other genes differentially expressed under drought stress related to hormone response as brassinosteroid and auxin-metabolism, which are relevant for plant growth and root development, respectively. We found that brassinosteroid-related genes, such as *Steroid 22-alpha-hydroxylase (DWF4)*, were downregulated, whereas genes involved in auxin-metabolism were both upregulated and downregulated (Appendix A).

### 2.6. Differentially Expressed Transcription Factors (TFs) in Tef under Drought Stress

A total of 734 transcription factors were differentially expressed in the Tsedey genotype under drought stress, of which 670 were upregulated while the remaining 64 were downregulated (Appendix A). The largest number of upregulated genes belonged to the Basic Leucine Zipper—bZIP (16%), Homeodomain-leucine zipper—*HD-ZIP* (14%), and Heat Shock transcription factor—HSF (7%) families (Appendix A). In contrast, a large number of downregulated TFs belong to the families of APETALA2/Ethylene-Responsive Factor—AP2-ERF (15%), WRKY (14%), and basic Helix-Loop-Helix bHLH (9%) (Appendix A). In addition, we examined the expression profiles of the top ten most differentially expressed transcription factors. Some of the upregulated genes encoding TFs were homeobox leucine zipper *HOX7*, bZIP transcription factor *TRAB1*, MYB transcription factor *ODOD1*, and heat stress transcription factor *A-2e* (Appendix A), whereas some of the downregulated genes encoding TFs were *ERF5*, *WRKY28*, and *Zinc Finger Protein Constants like 16* (Appendix A). We also identified genes involved in abiotic stress annotated by MapMan, such as heat stress transcription factors, heat shock proteins, germin-like proteins, peroxidases, and RAFTIN proteins (Appendix A). These genes might have a downstream position in the molecular network regulated by the transcription factors mentioned above.

### 2.7. Redox Related Genes Expression Is Modulated in Tef under Drought Stress

The production of enzymes for ROS scavenging is associated with plant adaptation to drought stress in plants. In this study, several differentially expressed genes under drought stress were found to be involved in redox reactions, including inorganic antioxidant enzymes such as glutaredoxin (GRX)*,* peroxidases (PRX), glutathione (GR), and thioredoxin (TRX), the majority of which were downregulated (Appendix A). In contrast, genes that play a key role in the synthesis of organic antioxidants including ascorbic acid and proline were upregulated under drought stress. In the ascorbic acid pathway, genes like *AAO* (*Ascorbate Oxidase*) and *DHAR1* (*Dehydroascorbate Reductase 1*) were upregulated (Appendix A). In proline biosynthesis, *P5CS (Δ^1^-Pyrroline-5-Carboxylate Synthetase*) was upregulated, while *glutamine synthase* was downregulated (Appendix A). Interestingly, both the upregulation of *P5C5* and downregulation of *glutamine synthase* boost proline synthesis. Moreover, proline biosynthesis enhances GABA shunt (Gamma-aminobutyric). Here, we found an upregulation of the *GABP* (*Gamma-aminobutyric Acid Transporter*) that facilitates the transport of GABA from the cytosol to the mitochondria (Appendix A).

### 2.8. Hypothesis of a Tradeoff Between Cell Wall-Related Genes and Starch Synthesis-Related Genes

The RNA-Seq results showed that there is likely a trade-off effect between the deacceleration of cell-wall biosynthesis proteins to accomplish the seed-filling stage (starch accumulation). Here, we showed that several genes involved in cell-wall biosynthesis, such as *Beta-xylosidase, Expansis, Pectinesterase*, and *Endo beta xylanase*, were downregulated (Appendix A). In contrast, we found genes involved in starch metabolism were upregulated, for example, beta-amylase enzymes. However, we also found events that could negatively affect starch accumulation, such as the downregulation of the *hexokinase 7* gene and the upregulation of *Trehalose phosphate phosphatase 9* (*TPP*) gene as it reduces the activation of AGPase (Appendix A). Further RNA-Seq analyses across different growth stages could help to clarify these results and to determine whether mature plants prioritize starch accumulation over cell-wall biosynthesis during drought stress. Appendix A shows the trade-off hypothesis where the activation of starch/sucrose metabolism (based on *Beta amylase* and *TPP* upregulation) might lead to a reduction of the cell wall metabolism, (i.e. less production of pectin and expansins).

### 2.9. RT-qPCR Validation of Differentially Regulated Transcripts

To assess the accuracy and reproducibility of the findings of the RNA-Seq, five differentially expressed genes in the RNA-Seq experiment were chosen for validation by the RT-qPCR. These five genes included four upregulated, namely *Ascorbic Acid Oxidase*-*AAO* (Et_5B_043714), *Δ^1^-Pyrroline-5-Carboxylate Synthase*-*P5CS* (Et_9A_061969), *Basic leucine Zipper transcription factor responsible for ABA regulation-bZIP TRAB1* (Et_2A_016154), and *Dehydroascorbate Reductase 1*-*DHAR1* (Et_9B_064015) and one downregulated, *Steroid 22-alpha-hydroxylase-DWF4* (Et_4B_038929) (Appendix A). As expected, a similar expression pattern was found for the five genes between the RNA-Seq and RT-qPCR. However, the expression levels were variable for the five genes. In this case, the RT-qPCR expression was increased 13-fold for *AAO,* 6-fold for *bZIP TRAB1*, 94-fold for *P5CS,* and 10-fold for *DHAR1* (Figure 4). Similarly, the transcript levels of *DWF4* were reduced 15-fold under drought stress.
Figure 4Validation of the RNA-Seq expression study by RT-qPCR. Relative gene expressions of five transcripts, namely, *AAO, bZIP TRAB1, P5C5, DHAR,* and *DWF4*, under drought conditions. Asterisks indicate significant differences using Student’s *t*-test to compare WW (well-watered) and WD (water-deficient) treatments. * = *p*  <  0.05; ** = *p*  <  0.01. Boxplots represent the mean ± SD. *n* = 3 biological replicates, where each replicate is a pool of five plants.
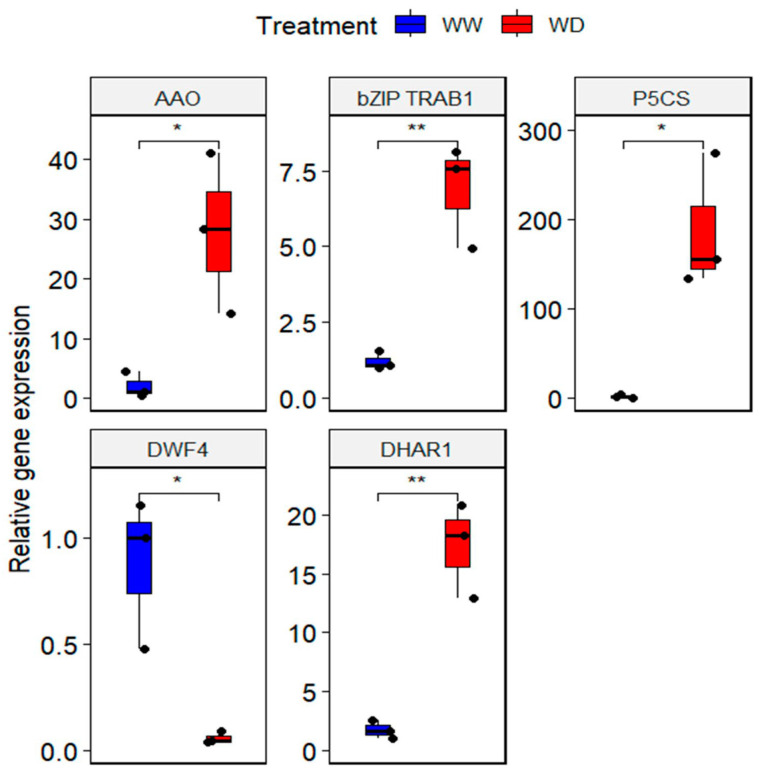


### 2.10. Proline Content in Tef Plants Exposed to Different Moisture Regimes

The synthesis of P5CS or *Δ^1^*-pyrroline-5-carboxlate enzyme is key in the proline synthesis pathway, which has properties related to redox balance and osmotic stress [31]. *P5CS* was upregulated under drought conditions in both the RNA-Seq and RT-qPCR experiments. Hence, to investigate the relationship between moisture scarcity and the abundance of proline in tef plants, leaf samples extracted from the *Tsedey* genotype under optimal watering and drought conditions were determined for proline content. The findings showed that the proline content increased 40-fold under drought stress compared to optimal water conditions (Figure 5).

## 3. Discussion

Tef has been classified as moderately drought-tolerant compared to other *Eragrostis* species [26]. However, a huge diversity exists in the response to different moisture regimes among the large number of tef genotypes, including natural accessions and farmers’ cultivars. [1,15,32,33].

Earlier studies showed that the tef genotypes *Kaye Murri*, *Ada*, and *Fesho* were more drought tolerant compared to *Balami* and *Alba* based on measurements of root length and osmotic adjustment [34]. The availability of the genome sequence of the *Tsedey* tef genotype [13] facilitates the current study, which also uses the same genotype. In the last decade, differential expression studies of transcriptomes, metabolomes, and proteomes elucidated the relationship between genotype and phenotype in various organisms. In this study, we offer the first data about the changes in the expression of the transcriptome of tef plants in response to water deficits.

### 3.1. Protein Receptors: Early Signals in the Plant Membrane

Signal transduction pathways enhance the responses to stimuli and signal transduction, which are necessary in abiotic/biotic stress responses. Receptor kinases (RLKs) are crucial for the signaling machinery, and they are also implicated in abiotic stress tolerance [35].

In the present study, the protein family with the most members downregulated was the Leucine-Rich Repeats (LRR) protein kinase, which is known to have functions in cell proliferation, stem cell maintenance, hormone perception, defense, and wounding responses [36,37]. This result is consistent with that of the differential gene expression analysis in little millet, where several genes encoding LRR receptors like serine/threonine protein kinases were downregulated due to salinity stress. Similarly, in rice and *Arabidopsis,* the *LRR-RLK* gene named *Panicle 2* (*LP2*), with a role in stomatal closure and density, was downregulated under drought and ABA exposure. Interestingly, members of the WAK (Wall-Associated Kinases) protein family, which are induced by abiotic and biotic stresses, were also downregulated [38,39]. Other genes encoding RLK protein members such as *S-RLK*, which is involved in the self-incompatibility response of Brassicaceae [40], and members of the L-lectin receptor protein family, involved in stress perception [41], were also downregulated under drought stress in tef. Finally, the plant-specific Domain of Unknown Function 26 family (DUF26), which is implicated in antifungal activity [36] has members that were repressed and activated under drought stress in tef.

### 3.2. Hormone Metabolism Activated by Drought Stress in Tef

Hormones play a crucial role in plant development and response to external stresses. Particularly, ABA signaling is known to regulate stomatal aperture to reduce water loss during moisture scarcity [42]. In the current study, genes encoding enzymes involved in ABA biosynthesis, such as *NCED (9 cis epoxycarotenoid dioxygenease NCED chloroplast), Abscisic Aldehyde Oxidase (AAO),* and *GT (UDP Glycosyltransferase protein*), were upregulated (Figure 3). Previous studies indicate that 9-*cis*-epoxycarotenoid dioxygenase (NCED), zeaxanthin epoxidase (ZEP), and aldehyde oxidase (AAO) are key enzymes in the ABA biosynthesis pathway in diverse plant species [43].

While the expression of *NCED* increases under drought stress in maize, the overexpression of *AtNCED3* enhances drought-inducible genes and decreases transpiration through ABA accumulation in *Arabidopsis* [44]. On the other hand, the overexpression of *AtZEP* in *Arabidopsis* increased ABA levels and reduced water loss in drought and high salinity tolerant genotypes [45]. Similarly, the AAO enzyme, which converts ABA aldehyde to ABA and catalyzes the final step of ABA biosynthesis, increased ABA production and reduced water loss in *OsAO3*-overexpressing rice lines, although the grain yield per plant was decreased in transgenic plants [46].

The current study revealed the differential expression of genes involved in brassinosteroids and auxins biosynthesis (Appendix A). Brassinosteroids (BRs) affect plant growth and development, particularly tiller number, leaf size, and leaf angle [47,48]. *DWF4* encodes a C-22 hydroxylase, and it is a key flux-determining enzyme that limits the endogenous level of BRs. The overexpression of maize *ZmDWF4* in *Arabidopsis* increased seed numbers under optimal growth conditions [49]. Similarly, transgenic rice plants overexpressing *sterol C-22 hydroxylases* from maize, rice, and *Arabidopsis* generated more seed tillers than wildtype plants under optimal growth conditions. These results suggest that BR stimulates carbohydrates assimilation from the source (leaves) to the sink (seeds) to achieve seed filling [47]. However, our results showed that *DWF4* was downregulated under drought stress, indicating that the growth of tef might be arrested. Earlier studies also showed that exogenous auxin application enhanced drought tolerance in plants [50,51]. In the current study, the *SAUR36* (*Small Auxin-Up RNA*) gene was downregulated (Appendix A), although the same protein is activated rapidly after auxin signaling and is related to the drought response [52]. The repression of auxin synthesis genes and the downregulation of brassinosteroid-related genes in the present study (Figure 3) are in agreement with the fact that auxin stimulates the production of brassiosteroids by increasing the expression of *DWF4* [53].

### 3.3. The Regulation of Transcription Factors is a Drought Response

Transcription factors are part of an early response to abiotic stresses and allow the activation of a complex molecular network [54,55]. In the current study, a high number of genes that are members of the *HD-ZIP, bZIP,* and *MYB TF*-families and *HSF transcription factors* were upregulated under drought stress in tef. Among the upregulated HD-ZIP transcription factors, *homeobox leucine zipper protein hox* (Et_1A_005486) was upregulated (Appendix A). Previous reports showed that HD-Zip genes were involved in abiotic stresses in foxtail millet [56] and wheat [57]. In this study, we found an upregulation of *bZIP TRAB1* (Et_2B_020259) (Appendix A), which is also known to be responsive to salt treatment in oats [58]. TRAB1 also activates the expression of ABA responsive genes in rice and barley [59,60]. Interestingly, the OsbZIP23 TF in rice shares a high sequence similarity with the OsTRAB1 target genes involved in drought tolerance response. This includes the activation of *OsNCED4*, a key gene of ABA signaling, by binding to the promoter region [61]. Therefore, it will be interesting to see whether an overexpression of *TRAB1* in tef under drought stress activates ABA responsive genes. Finally, in our study, the third most differentially upregulated transcription factor family was the MYB family. The study on oats also showed *MYB transcription factors* were upregulated under salt stress [62].

Interestingly, a high proportion of members of the *AP2, WRKY*, and *bHLH* transcription factor families were downregulated under drought in the present study (Appendix A). Although the overexpression of *AP2* is usually related to drought tolerance responses [63,64], in sorghum, the repression of *AP2* (*SOBIC.002G071600*) promotes drought tolerance in response to polyethylene glycol (PEG) treatment [65]. On the other hand, genes encoding WRKY and bHLH transcription factors were regulated in response to drought and salinity stress in little millet [66].

Although these results give a global view of the high number of transcription factor families regulated in tef under drought stress, validating TFs that are differentially expressed under drought is necessary. Similarly, the identification of downstream genes regulated by the above-mentioned transcription factors and regulatory transcription factors involved in crosstalk under abiotic stress is crucial for understanding plant adaptation to stresses.

### 3.4. Abiotic Stress-Responsive Genes

Heat Shock Proteins (HSPs) were also upregulated under drought stress (Appendix A). HSFs proteins control *HSPs*’ expressions through binding to the promoter region [67]. HSPs are chaperones that play a role in folding and activating proteins involved in signal transduction. Earlier studies in pearl millet found that genes encoding HSPs were differentially expressed under drought and heat stress [68]. In soybean, heat shock proteins were expressed as an early response to flooding and drought stress [69,70]. In the future, an investigation of the expression of HSPs in tef plants exposed to mild drought treatment is warranted.

Conversely, genes encodingGermin-like proteins (GLPs) were downregulated under drought stress in the present study involving tef (Appendix A). Earlier studies in rice showed that GLPs were involved in abiotic stress tolerance responses due to their antioxidant activity [71]. Members of the GLP family possess numerous motifs of AP2/ERFbs transcription factors in their promoter regions, indicating that they play a regulatory role in abiotic stress response [71].

### 3.5. Antioxidant Activity Modulated During Drought Response

Although several inorganic antioxidants were repressed during drought stress, synthesis of organic antioxidants such as proline and ascorbic acid were activated in the current study (Appendix A). During the stress, the antioxidant activity increased in tolerant plants. Ascorbic acid (AsA) is a non-enzymatic ubiquitous antioxidant with the potential for scavenging ROS and regulating biological functions in plants, especially under stress conditions [72]. The exogenous application of AsA in wheat mitigated salinity stress by increasing chlorophyll, carotenoids, proline accumulation, and leaf area while decreasing H_2_O_2_ levels in plant tissues [73].

In the present work, *APX, AO,* and *DHA* genes were upregulated in tef plants subjected to drought (Appendix A). APX, AO, and DHA are enzymes involved in the recycling AsA system that act by protecting cells from oxidative stress [74]. The overexpression of *AgAPX1* from celery in *Arabidopsis* showed a drought-tolerant phenotype with a higher antioxidant capacity [75]. Similarly, the knockdown of *APX4* in rice showed an early leaf senescence under optimal growth conditions [76]. Moreover, the overexpression of *Arabidopsis* cytosolic *DHAR* and *MDHAR* genes lead to a higher level of AsA in transgenic tobacco plants exposed to diverse abiotic stresses including aluminum, salinity, and drought [77,78].

The current work also showed an increase in proline content (Figure 5) together with upregulation of *P5CS* (Figure 4)—a key enzyme in proline biosynthesis that catalyzes the reaction from glutamate to P5C (Δ¹-pyrroline-5-carboxylate) and acts in the latest reactions to synthesize proline. Proline is a multi-functional amino acid that serves as an osmoprotectant during water limitation, and it has a role in redox buffering [79]. The *p5cs1* knockout mutant of *Arabidopsis* showed reduced levels of proline during low water potential [80]. The ectopic expression of *P5CS* from beans increased stress tolerance in wheat in a way that is associated with higher proline content [81]. The same study showed that *P5CS*-transformed plants had a reduction in free radical levels during water withholding, indicating its role as an antioxidant. Similarly, tobacco plants transformed with *P5CS* had higher proline accumulation and lower MDA content in response to freezing stress [82]. In addition, Glutamine Synthase (GS), which catalyzes the reaction from glutamate into glutamine, was downregulated (Appendix A). Previous work showed that the *GS2* mutant in *Lotus japonicus* had a lower proline accumulation than non-transformed plants under drought [83]. Therefore, it is possible that under drought stress, the demand for proline synthesis is higher, which could prioritize the production of glutamate rather than glutamine.

Taken together, the overexpression of key genes is vital in developing drought tolerant tef lines. Hence, we propose further studies imposing drought stress during short- and long-periods to determine whether genes encoding organic and inorganic antioxidants are differentially expressed in tef.

### 3.6. Hypothesis of Tradeoff Between the Cell Wall and Starch Metabolism

Glucose in plants can be redirected into different metabolic pathways depending on cellular needs, particularly under environmental stresses. Our results showed a possible tradeoff between the cell wall and starch metabolism, where most of the cell wall-related genes were downregulated (Appendix A). In the case of starch metabolism targets, genes were both upregulated and downregulated. We hypothesize that the tef *Tsedey* genotype prioritizes starch assimilation during drought stress to accelerate or ensure plant growth and seed development.

Regarding starch-related genes, we observed a decrease in the expression of the gene *hexokinase 7* (Appendix A), which encodes one of the first enzymes that catalyzes starch synthesis from glucose 6 phosphate to glucose. Previous studies showed that enhancing the expression of both *SP6A* and *AtHXK1* in potato improves water efficiency and minimizes yield loss under heat and drought stress [84]. These results suggest that a reduction of hexokinase expression would affect yield during drought stress.

Interestingly, we also found that *Trehalose-Phosphate Phosphatase 9* (*TPP*), a gene encoding an enzyme that catalyzes the conversion of trehalose-6-phosphate (T6P) to trehalose, was upregulated (Appendix A). T6P is involved in the AGPase activation that facilitates starch accumulation. In contrast, TPP converts T6P into trehalose. In *Arabidopsis*, plants overexpressing *TPP* decreased their redox activity and had less activation of AGPase and reduced starch content [85]. However, it is well-known that trehalose, the product of the TPP enzyme, is a protective molecule that helps plants cope with various stress conditions, acting as an osmoprotectant in cells to maintain cellular integrity [86].

A decrease in starch accumulation affects crop yield during drought conditions [87]. Further, beta-amylase enzymes break down starch into maltose [88]. This study found an upregulation of genes encoding *beta-amylase* enzymes (Appendix A), located in the amyloplast, which could indicate an important activity in the latest step of starch metabolism. During both severe and moderate drought, alpha and beta amylase enzymes are boosted in cassava [89]. Studies performed on rice at the seedling stage under anoxic conditions increased the expression of the *Amy3* subfamily gene, likely because of the lack of sugar [90]. Moreover, the overexpression of *VvBAM1* (*Vitis vinifera beta amylase 1*) in tomato not only improved cold tolerance but also facilitated starch breakdown and mitigated the production of reactive oxygen species [91].

Based on the current study, we hypothesize a tradeoff in which the accumulation of starch in tef is being compensated for by a disruption in cell wall biosynthesis. Plant cell expansion and remodeling are key factors in regulating internal turgor pressure inside the cell, especially under stress [92]. Here, we found a downregulation of genes encoding expansin proteins under drought stress (Appendix A). Expansins are involved in cell wall extension and maintaining turgor pressure under water scarcity. The overexpression of *expansin* (*EXLA2*) in *Arabidopsis* increased the hypersensitivity of the plant to salinity and cold stress [93]. Similarly, the overexpression of the expansin-like gene *GhEXLB2* enhanced drought tolerance in cotton by increasing the activities of peroxidase and superoxide dismutase [94]. Importantly, expansins are also involved in plant hormone induction such as ABA, auxin, and brassionsteroids [95,96,97]. For instance, auxin positively regulates expansins and promotes cell wall loosening (relaxation) to favor plant growth [98]. In addition, genes encoding xyloglucan endotransglucosylase/hydrolase enzymes were also downregulated; plant cell walls are compounds of hemicellulose, pectins, and cellulose proteins where xyloglucan is the prevalent hemicellulose. The xyloglucan endotransferase/hydrolase (XTH) enzymes play a role in modifying the fiber-xyloglucan complex, hence affecting cell wall remodeling. In several species, it has been demonstrated that the overexpression of these genes increase multiple abiotic stress tolerance. For example, the overexpression of poplar *PeXTH* enhanced salt and cadmium resistance in tobacco [99]. Similarly, the overexpression of soybean *GmXTH23* in *Arabidopsis* promoted root development and drought tolerance [100]. Therefore, it is possible that the overexpression of *XTH* in tef might increase drought tolerance in the crop.

Further, we found that genes encoding pectinesterase inhibitors 12 enzymes were downregulated (Appendix A). Pectins are major cell wall matrix components that also facilitate cell wall plasticity. The pectinesterase enzyme catalyzes the hydrolytic cleavage of methyl ester moieties on pectin molecules, releasing methanol and partially de-esterified pectin [101]. De-esterified pectin leads to a less compact cell wall structure, while the pectinesterase inhibitor hampers with pectinesterase activity, leading to a firmer cell wall [102]. Moreover, a study in soybeans found that *pectinesterase inhibitor* was downregulated in a tolerant genotype and upregulated in a sensitive genotype [103]. Therefore, it is possible that pectinesterase inhibitors 12 may have the opposite effect of the expansin proteins during drought stress. However, further biological studies at the functional level need to be conducted to confirm this hypothesis.

Therefore, a possible balance between cell wall and starch metabolism in tef can help the plant to overcome drought stress. Further studies identifying differential gene expression not only in leaf tissue but also using reproductive structures could provide valuable insights into the relocation dynamics of sugars derived from the beta-amylase-mediated starch breakdown in tef during drought stress.

Based on the findings, we showed how tef plants regulate a complex molecular network involving membrane receptors and transcription factors that can be ABA-dependent or ABA-independent (Figure 6). We also highlighted the binding motifs that each transcription factor recognizes in the promoter region of target genes to modulate their expression. Furthermore, we show plant responses through the regulation of hormones such as ABA (stomatal regulation), auxin (root development), and brassinosteriods (plant growth). Additionally, we highlight the activation or repression of antioxidants, osmoprotectants, and cell wall and starch metabolism genes. Engineered plants based on the overexpression or downregulation of these candidate genes will help to elucidate their biological functions. Since the current study was only conducted on *Tsedey*, an improved tef variety adapted to semi-arid areas in Ethiopia, it is important to investigate the response of representative genotypes of diverse agroecological zones and stress conditions. The availability of these genotypes would be key in identifying polymorphisms that might lead to drought tolerance. Polymorphisms in the promoter region might affect the binding of transcription factor(s) to the upstream region of the target gene. This information is also valuable for transferring knowledge to other abiotic stresses, given the known crosstalk among different stress responses. Finally, as an ancient crop capable of growing in diverse geographic locations and climatic conditions, tef may possess genetic information that enables its tolerance to diverse abiotic stresses in a better way compared to cereals such as maize and rice. Therefore, a future scenario will be to enhance yields in these economically important cereals by regulating key genes or identifying polymorphisms that boost drought tolerance.

## 4. Materials and Methods

### 4.1. Plant Material and Experimental Setup

Plants of the improved tef variety *Tsedey* (also known as DZ-Cr-37) were grown in pots under long-day conditions (16 h light at 22 °C and 8 h dark at 18 °C), a relative humidity of 50%, and a light intensity of 170 mmol/m^2^/s photosynthetically active radiation at the plant level. After 19 days of optimal watering, either watering was continued (control), or water was withheld for 9 days. The soil moisture content of plants exposed to 9 days of drought was 7%, whereas the control plants had 70% according to the TDR/MUX/mpts soil moisture probe device.

### 4.2. Library Construction and RNA Sequencing

The RNA extracted from plants grown under water deficit and normal watering conditions was sent to Fasteris (Geneva, Switzerland) for further quality testing and sequencing using Illumina HiSeq2000 (San Diego, CA, USA). Two biological replicates were collected from the leaves of the control plants and those subjected to drought, resulting in two libraries each for control (GNY1, GNY10) and drought (GNY2, GNY11). The quality and quantity of RNA were quantified using an ND-1000 Spectrophotometer (Thermo Fisher Scientific, Waltham, MA, USA), and the average 260/280 ratio was at least 2.0, indicating good-quality RNA. The GNY1a (control) and GNY2a (drought) libraries were prepared with a TruSeq SBS v5 kit, and a data analysis pipeline consisting of HiSeq Control Software version 1.1.37.8, RTA 1.7.48, and CASAVA 1.7 was used. GNY1b, GNY2b, GNY10, and GNY11 also used the TruSeq SBS v5 kit (Illumina, San Diego, CA, USA) and flow-cell version 3 with the following software: HiSeq Control Software version 1.4.8, RTA 1.12.4.2, and CASAVA 1.8.2. The GNY10 (control) and GNY11 (drought) libraries were prepared using AccuPrime™ *Taq* DNA Polymerase System (Invitrogen, Carlsbad, CA, USA) following the protocol for high GC content. The six cDNA libraries were sequenced to generate about 134 million single-end reads. Before assembly and mapping, the reads were trimmed such that the Phred quality scores were above 28. In addition, all primer and adaptor sequences detected by FastQC were removed.

### 4.3. Analysis of Differentially Expressed Genes (DEGs)

The change in transcript expression between the drought and control conditions was determined as follows: the reads from each condition were mapped onto the 14,057 scaffolds of size 1000 or greater obtained from The Tef Improvement Project [13] using STAR 2.3.0 [104] with the default parameters. These aligned reads were converted to the BAM format with SAMtools [105]. A count table was obtained using the HTSeq-count program with options stranded = no, type = gene, and attribute = ID [27] and using the Maker gene predictions provided with the tef genome [13]. HTSeq reported the percentages of reads mapped, reads mapped to unique locations, reads mapped to multiple locations, and unmapped reads. Only the reads that mapped uniquely to one location were used to generate count tables. To increase confidence, significant DEGs were identified using the false discovery rate (FDR) ≤ 0.01 and |log2FoldChange| ≥ 2.

### 4.4. Annotation and Enrichment Analysis

The Gene Ontology (GO) enrichment analysis was implemented using the topGO R package, using the Dabbi tef GO term annotation as a reference [14]. A Fisher’s exact test with the weight algorithm was implemented in topGO [106] with a nodeSize set to 10 for all the GO enrichment analyses. Upregulated and downregulated genes were classified separately intro three major categories: biological process (BP), cellular component (CC), and molecular function (MF).

MapMan Mercator software was used to match the annotation information of the gene sequences of tef. The parameters were a BLAST cutoff of 80 with multiple bin assignments allowed. The database sequences used were from TAIR, SwissProt/UniProt Plant Proteins, TIGR5 rice proteins, and the KOG database. Results mapping from Mercator were used to visualize functions of the differentially expressed genes in MapMan [30].

### 4.5. Physiological Measurements

After the end of the water stress period and before harvesting leaf samples, stomatal conductance of the adaxial side of the flag leaf was determined using an AP4 diffusion porometer (Delta T, Cambridge Life Sciences, Cambridge, UK) using ten biological replicates. Chlorophyll a and b as well as carotenoids (carotenes and xanthophylls) were extracted using 95% ethanol and measured with UV-Vis Spectroscopy [107]. The measurements were repeated on ten biological replicates. The amount of pigment was normalized by fresh weight. Significant differences between well-watered and drought treatments were tested with a Student’s *t*-test, using a *p*-value of ≤0.05 to determine statistical significance between treatment means. Finally, to measure the relative water content (%), we used the second leaf from the top as a sample, and the formula was calculated as follows: “RWC (%) = [(FW − DW)/(TW − DW)] × 100”, where FW is fresh weight, DW is dry weight, and TW is turgid weight of the sample material. Five biological replicates were used for this measurement.

### 4.6. Validating the Findings of Gene Expression

Total RNA was extracted from tef leaves exposed to drought or normal watering using the Total RNA Isolation System (Promega) and treated with DNase I (Promega) to remove the DNA in the samples. The concentration of the total RNA was adjusted to ~0.1 μg/μL. The first-strand cDNA synthesis was performed using M-MLV reverse transcriptase (Promega, Madison, WI, USA), while the second-strand synthesis was conducted in the LightCycler^®^ 96 System (Roche, Basel, Switzerland) using the FastStart Essential DNA Green Master Kit (Roche, Basel, Switzerland), following the program: 95 °C for 10 s; 60 °C for 10 s; 72 °C for 12 s for 45–55 cycles. The relative gene expression was quantified using the 2^−△△Cq^ method [108].

Five differentially expressed genes from the RNA-Seq were validated by RT-qPCR using *α-Tubulin 1* as a reference gene. The list of the primers is shown in Appendix A. An unpaired two-tailed Student’s t-test was performed to determine the significant differences between the control and drought treatments.

### 4.7. Quantifying Proline Content

Proline content was measured at the end of the water withholding treatment at the vegetative stage of the tef plant. A 100 mg sample was taken for each treatment from the flag leaf where five biological replicates were used per treatment. The ninhydrin protocol was used to determine proline content [109]. The frozen powder of the leaf sample was added to 1.8 mL of 3% sulfosalicylic acid and mixed by vortexing and incubating on ice for 30 min followed by spinning at 10,000 rpm for 5 min. The supernatant was transferred to a glass tube with 300 μL acetic acid and 300 μL ninhydrin reagent. The ninhydrin reagent for each reaction consists of 7.8 mg ninhydrin, 187.5 μL glacial acetic acid, and 125 μL 6M phosphoric acid, which were mixed at 50 °C. Samples were incubated for 1 h in a boiling water bath, briefly cooled on ice, and 1.5 mL toluene was added and vortexed. The absorbance of the upper layer was measured at 520 nm in a glass cuvette. Proline concentrations were determined by comparing the results to a standard curve generated from serial dilutions of proline stock solutions.

## Figures and Tables

**Figure 1 plants-13-03086-f001:**
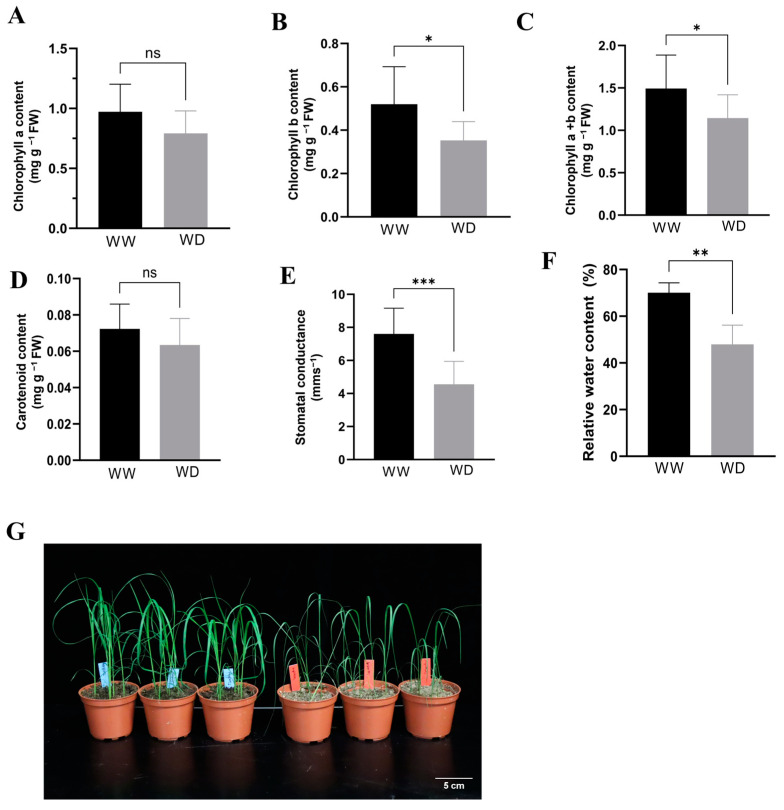
Effects of withholding water for nine days on physiological parameters of tef plants: (**A**) chlorophyll a; (**B**) chlorophyll b; (**C**) total chlorophyll (a + b); (**D**) carotenoid content; (**E**) stomatal conductance; (**F**) relative water content (%); (**G**) visual phenotypes of well-watered (left) and drought affected (right) plants. Bars represent the mean ± SD. * = *p* < 0.05, ** = *p* < 0.01 and *** = *p* < 0.001. WW: well-watered; WD: water-deficient.

**Figure 2 plants-13-03086-f002:**
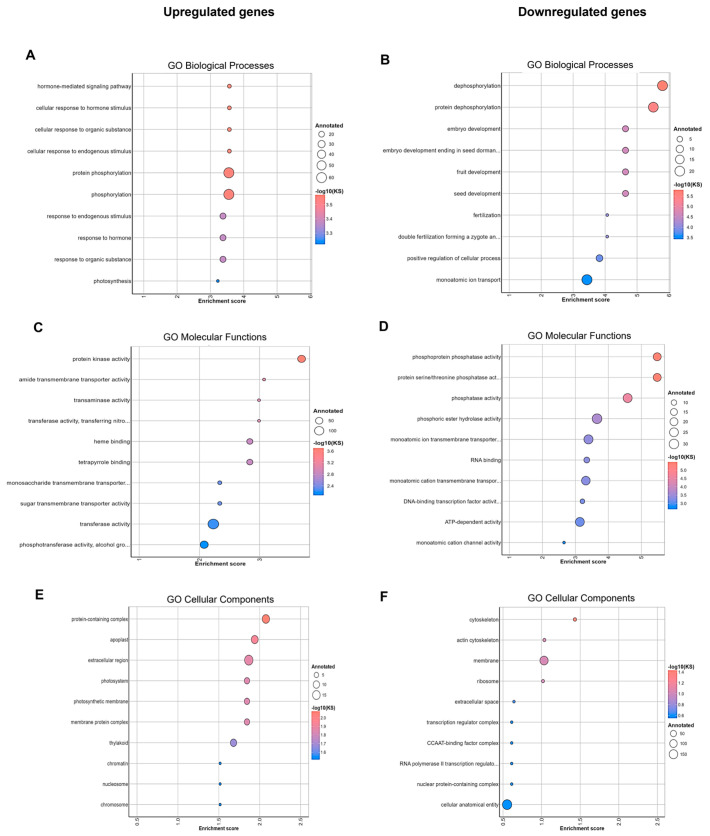
GO term enrichment of genes upregulated and downregulated under drought conditions. TopGO was used to provide functional annotations of the sets of genes significantly upregulated and downregulated in three broad categories: biological processes (**A**,**B**), molecular functions (**C**,**D**), and cellular component (**E**,**F**). The size of the circles represents the number of genes annotated under that GO term in the genome, and the color indicates −log10 (*p*-value) of DEGs annotated in the specific GO analysis. Terms with a *p*-value ≤ 0.05 included in the graph are sorted from lowest to the highest significant *p*-value. The *p*-values were obtained by using the KS test where KS represents the value from the Kolmogorov–Smirnov test used to determine whether the GO enrichment was significant, i.e., KS ≤  0.05.

**Figure 3 plants-13-03086-f003:**
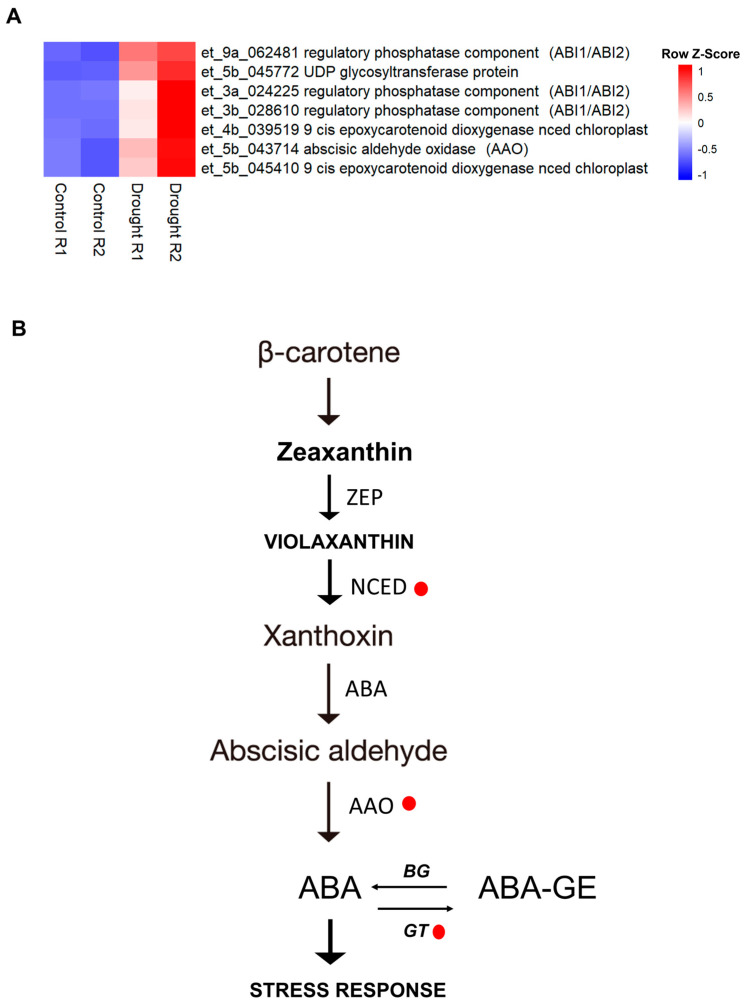
Expression pattern of genes involved in the ABA pathway in tef under drought stress. (**A**) Overview of ABA-related genes differentially regulated in *Tsedey* under drought stress. Differentially expressed genes (fold change > 2, *p*-value < 0.05) are represented as red (upregulated) and blue (downregulated) squares. (**B**) ABA pathway. The red circles represent the genes upregulated in tef under drought stress. AAO: aldehyde oxidase, ZEP: zeaxanthin epoxidase, NCED: 9-cis-epoxycarotenoid dioxygenase, ABA: Abscisic acid, GT: glucosyltransferase, BG: β-glucosidases. The complete list of genes involved in ABA biosynthesis is available in Appendix A. The expression of AAO was validated by RT-qPCR in the current study (shown in Figure 4).

**Figure 5 plants-13-03086-f005:**
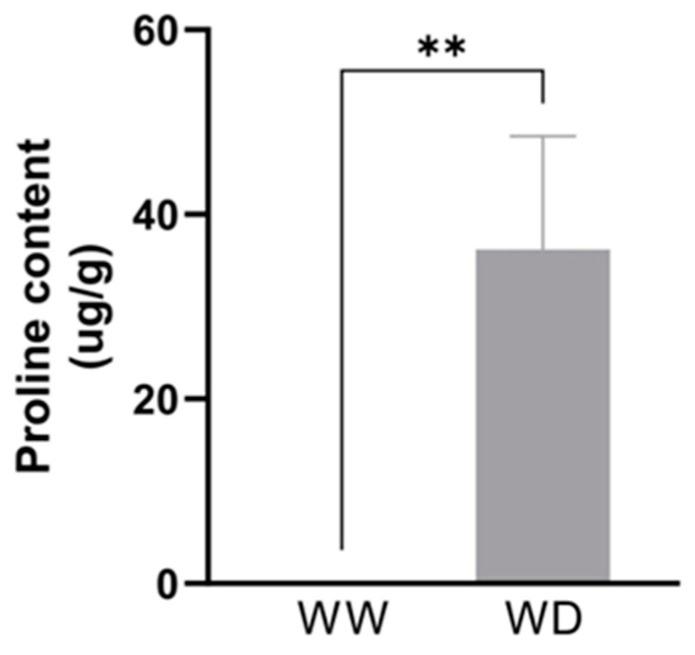
Proline content (ug/g sample) of *Tsedey* subjected to drought conditions. *Tsedey*: *n* = 5. Bars represent the mean ± SD. ** = *p* < 0.01. WW: well-watered; WD: water-deficient.

**Figure 6 plants-13-03086-f006:**
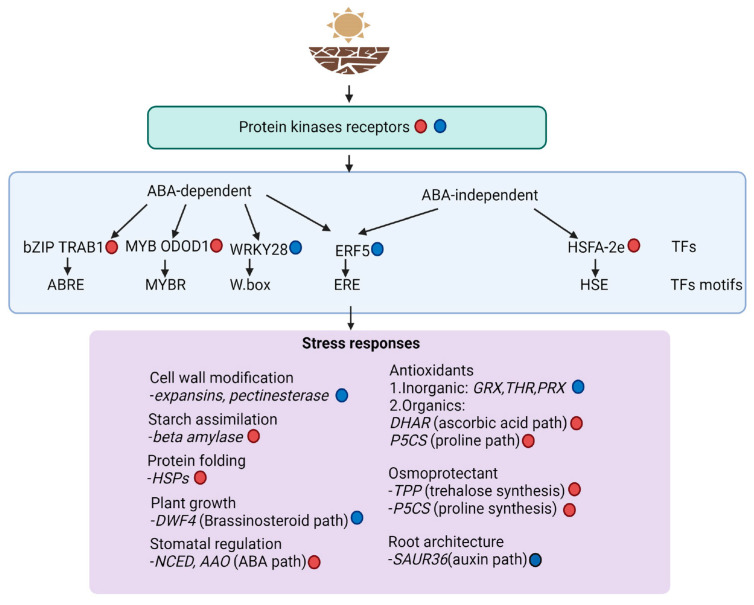
Schematic diagram showing differential gene expression during drought stress. The molecular networking begins with the regulation of cell membrane receptors, followed by the modulation of transcription factors (highlighting the binding motifs they recognized) and finally the stress responses showing key genes involved in pathways of cell wall modification, starch assimilation, plant growth, stomatal regulation, antioxidants, osmoprotection, protein folding, and root architecture. The red and blue circles represent genes that were upregulated and downregulated, respectively, in the RNA-Seq experiment.

## Data Availability

Data in this project have been archived at Genbank under BioProject PRJNA413657 with BioSamples: GNY1: SAMN07764637, GNY2:SAMN07764638, GNY10:SAMN07764640 and GNY11: SAMN07764641. The reads have been deposited in the Genbank Sequence Read Archive as study SRP119988 with the following accession numbers: SRR6175533 (GNY1-1), SRR6175534 (GNY2-1), SRR6175535 (GNY1-2), SRR6175529 (GNY2-2), SRR6175530 (GNY10) and SRR6175532 (GNY11).

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
