# Peer review of "Transcriptomic Profile of Tef (Eragrostis tef) in Response to Drought"

_plants, 2024, doi:10.3390/plants13213086_

Round 1

Reviewer 1 Report

Comments and Suggestions for Authors

Dear Authors,

the manuscript “Transcriptomic profile of tef (Eragrostis tef) in response to drought” is well written, in a clear and detailed manner it reports interesting information relating to the genes differentially expressed in response to water stress in Eragrostis tef.

However, I found some inaccuracies which I report below:

Line 216 The phrase “Differentially expressed genes (fold change > 2, p-value < 0.05) are represented as red (down-regulated) and blue (up-regulated) squares” was reported in the captions of all the figures, even those supplementary, which refer to heatmaps, but in my opinion they lead to confusion in relation to the up and down regulated genes compared to what is reported in the rest of the caption and also in the text of the manuscript. I suggest checking its correctness and possibly making it more consistent with what is observed in the figures.

Line 267 I don't see Figure 4 relating to the RT-qPCR validation of the expression profiles of 5 genes obtained from the transcriptome analysis.

Line 274 Please, correct the word “Pfigure5CS” to “P5C5”.

Line 565 Table S1 reports “Description of oligo primers used in the RT-qPCR experiment to validate differentially expressed genes in the RNA-Seq study” and not information regarding the cDNA libraries, therefore the table referred to in the text is missing.

Author Response

Comments 1: [Line 216 The phrase “Differentially expressed genes (fold change > 2, p-value < 0.05) are represented as red (down-regulated) and blue (up-regulated) squares” was reported in the captions of all the figures, even those supplementary, which refer to heatmaps, but in my opinion they lead to confusion in relation to the up and down regulated genes compared to what is reported in the rest of the caption and also in the text of the manuscript. I suggest checking its correctness and possibly making it more consistent with what is observed in the figures.]

Response 1: [The symbols are now consistent where by red represents up-regulated while blue represents down regulated.] Thank you for pointing this out.

Comments 2: [Line 267 I don't see Figure 4 relating to the RT-qPCR validation of the expression profiles of 5 genes obtained from the transcriptome analysis.Paste the full comment here.]

Response 2: [we have inserted the correct Figure 4. Sorry for the mistake.]

Comments 3: [Line 274 Please, correct the word “Pfigure5CS” to “P5C5”.]

Response 3: [we corrected the word P5C5.]

Comments 4: [Line 565 Table S1 reports “Description of oligo primers used in the RT-qPCR experiment to validate differentially expressed genes in the RNA-Seq study” and not information regarding the cDNA libraries, therefore the table referred to in the text is missing.]

Response 4: [we cited Table S1 in the right place in the manuscript.]

Reviewer 2 Report

Comments and Suggestions for Authors

This study presents the results of RNA-Seq analysis to identify a complex molecular network involving membrane receptors and transcription factors that regulate drought responses in the genome of Tef (Eragrostis tef) an allotetraploid cereal crop.
The ultimate goal, identification of genes for response to drought stress.

The work is generally descriptive, based on known protocols, like many similar works.
It is extremely difficult to analyse such work or to verify the validity of the results obtained. The complexity of RNA-Seq analysis for phenotype contrasting samples is related to the large amount of data, in which correlation between gene expression for the compared samples is not necessarily found.
The authors use qPCR to validate the RNA-Seq results obtained. However, this approach does not guarantee the reliability of the obtained data.
Technically, the qPCR method is preferably used as multiplex PCR for simultaneous detection of different genes and for different sites of target genes, using TaqMan probes. There is no such analysis in this study.
In addition, more genes should also be included in the study, for control analysis.
However, even when confirming contrasting transcription for the target gene in genetically contrasting genotypes, it does not indicate that protein synthesis will also be increased under stress conditions.
This is a major problem in such studies, and in my opinion it is preferable to study the protein component instead of qPCR analysis.

Could not access the data:
Supplementary 1
Doi: 10.5281/zenodo.13836945 ; Link: https://zenodo.org/records/13836945?preview=1&token=eyJhbGciOiJIUzUxMi

Author Response

Comments 1: [It is extremely difficult to analyse such work or to verify the validity of the results obtained. The complexity of RNA-Seq analysis for phenotype contrasting samples is related to the large amount of data, in which correlation between gene expression for the compared samples is not necessarily found. The authors use qPCR to validate the RNA-Seq results obtained. However, this approach does not guarantee the reliability of the obtained data. Technically, the qPCR method is preferably used as multiplex PCR for simultaneous detection of different genes and for different sites of target genes, using TaqMan probes. There is no such analysis in this study. This is a major problem in such studies, and in my opinion it is preferable to study the protein component instead of qPCR analysis.]

Response 1: [We used qPCR, as a first step, to validate the findings of RNA-Seq. As you have indicated, protein analysis can be done as second stage. As protein analysis requires additional resources, time and expertise, we believe that the validation with qPCR is suffice for this exploratory study as research on this understudied crop receives little funding.]

Comments 2: [In addition, more genes should also be included in the study, for control analysis. However, even when confirming contrasting transcription for the target gene in genetically contrasting genotypes, it does not indicate that protein synthesis will also be increased under stress conditions.]

Comments 3: [Could not access the data:
Supplementary 1. Doi: 10.5281/zenodo.13836945 ;

Link: https://zenodo.org/records/13836945?preview=1&token=eyJhbGciOiJIUzUxMi ]

Response 3: [The correct addresses of Zenodo and doi are indicated below. The same information are also included in the legend of supplementary materials in the main manuscript.

Link: https://zenodo.org/records/13836945?preview=1&token=eyJhbGciOiJIUzUxMiJ9.eyJpZCI6ImQyNWZiYzJlLWI5ZGMtNDYyMS05NDFjLWQwZWZlMjY0NjYzNSIsImRhdGEiOnt9LCJyYW5kb20iOiJkOGQ3MTFlMzRjNDU2Njg4NGZiYjgxNzA0MmUwYTMxNCJ9.fIeSLbUKB1klgZCRa55zkHp6JJmjTHNGg0w_9NFVesVX67xUv3pa60JFbylTle_gZ2yWTjqf_Gqxt4eeAOf72Q

DOI: 10.5281/zenodo.13836945 ]